# Patient-Reported Outcomes (PROs) and Health-Related Quality of Life (HR-QoL) in Patients with Ovarian Cancer: What Is Different Compared to Healthy Women?

**DOI:** 10.3390/cancers13040631

**Published:** 2021-02-05

**Authors:** Melisa Guelhan Inci, Rolf Richter, Kathrin Heise, Ricarda Dukatz, Hannah Woopen, Jalid Sehouli

**Affiliations:** Department of Gynecology with Center for Oncological Surgery, European Competence Center for Ovarian Cancer, Campus Virchow Klinikum, Charité—Universitätsmedizin Berlin, Corporate Member of Freie Universität Berlin, Humboldt-Universität zu Berlin, and Berlin Institute of Health, Augustenburger Platz 1, 13353 Berlin, Germany; Rolf.richter@charite.de (R.R.); kathrin.heise@charite.de (K.H.); ricarda.dukatz@charite.de (R.D.); hannah.woopen@charite.de (H.W.); jalid.sehouli@charite.de (J.S.)

**Keywords:** patient-reported outcomes (PROs), quality of life (QoL), health-related quality of life (HR-QoL), ovarian cancer, elderly

## Abstract

**Simple Summary:**

The aim of this analysis was to evaluate the health-related quality of life (HR-QoL) in patients with ovarian cancer using a questionnaire based on patient-reported outcomes (PROs)**.** The HR-QoL for 155 enrolled patients with ovarian cancer was assessed by the European Organization for Research and Treatment of Cancer Quality of Life Questionnaire (EORTC QLQ-C30) prior to surgery and compared with 501 healthy females in Germany, as well as to the previously published European Organization for Research and Treatment of Cancer (EORTC) reference data for 917 patients with ovarian cancer worldwide. The HR-QoL for the emotional, cognitive, and social functioning scales was lower in patients with ovarian cancer than the healthy female population. Interestingly, the patients with ovarian cancer had no significant differences in the physical functioning scale when compared with the healthy women. Furthermore, the younger patients with ovarian cancer had an even lower HR-QoL for the emotional, social, and cognitive functioning scales, and additionally had more fatigue and financial difficulties.

**Abstract:**

Introduction: The aim of this analysis was to evaluate the health-related quality of life (HR-QoL) in patients with ovarian cancer using a patient-reported outcome (PRO) based questionnaire and to compare it to the healthy female population in Germany and to other ovarian cancer patients worldwide. Additionally, we looked for differences in the HR-QoL with respect to the patients’ ages in our cohort. Methods: The HR-QoL for 155 enrolled patients with ovarian cancer was assessed by the European Organization for Research and Treatment of Cancer Quality of Life Questionnaire (EORTC QLQ-C30) prior to surgery and then compared with 501 healthy females in Germany, as well as to the previously published European Organization for Research and Treatment of Cancer (EORTC) reference data for 917 patients with ovarian cancer worldwide. Moreover, we grouped our cohort by ages <65 and >65 years and analyzed them for further differences. To identify the differences, T-tests were applied. Results: Overall, 155 patients were enrolled, and 126 patients had advanced-stage ovarian cancer (FIGO III–IV) (82.4%). Fifty-five (36%) patients were >65 years. Except for the physical functioning scale, all other domains of the functioning scales were significantly lower in our patients with ovarian cancer than in the healthy female population. The emotional (50 points versus 60 points, *p* = 0.02), cognitive (76 points versus 88 points, *p* = 0.005), and social functioning scales (68 points versus 81 points, *p* = 0.006) were lower in the younger subgroup. Further, the younger subgroup exhibited significantly more fatigue (40 points versus 29 points, *p* = 0.03) and financial difficulties (20 points versus 2 points, *p* < 0.001) than the older subgroup. Discussion: Interestingly, the patients with ovarian cancer had no significant differences in the physical functioning scale when compared with the healthy women. In contrast, the patients, especially in the younger group, needed special support for the emotional and social areas of their daily lives.

## 1. Introduction

Ovarian cancer and its treatment affect all dimensions of a patient’s life: not only in regard to physical changes, but also the psychological and social aspects that affect the health-related quality of life (HR-QoL) of these women [1,2,3,4]. The required complex surgery, systemic therapy, and symptoms of an advanced-stage tumor or chronic disease can worsen their HR-QoL [5]. In patient-oriented care, we need patient-reported outcome (PRO) measurements, particularly to assess the quality of life. In order to be able to guarantee patient-oriented medicine, we have to know objectively what limitations patients with ovarian cancer have in their daily lives. The challenge here is to quantify a subjective judgment in an objective way that is both accountable and comparable. Therefore, assessments of various aspects of the patients’ quality of life and patient-reported outcomes are of special relevance.

An established tool to measure the quality of life in cancer patients is the European Organization for Research and Treatment of Cancer Quality of Life Questionnaire (EORTC QLQ-C30) [6,7]. This questionnaire is designed to measure physical, role, cognitive, emotional, and social functioning as well as certain symptoms, such as fatigue, pain, and nausea/vomiting. It is available in more than 100 languages, is used worldwide, has been validated in multiple studies, and accurately shows the patients’ perceptions [6,7,8,9]. Reference data for the Quality of Life Questionnaire (QLQ-C30) for ovarian cancer provided by the European Organization for Research and Treatment of Cancer (EORTC) Quality of Life Group was published in July 2008. In 2019, 15,000 people worldwide were asked to answer a EORTC-QLQ-C30 to establish a healthy reference collective [10].

Previous works demonstrated a strong association between the physical part of the EORTC QLQ-C30 and severe postoperative complications [11,12,13]. In particular, functioning domains such as low physical ability and some specific symptoms, most notably, pain, fatigue, and loss of appetite, were associated with poorer survival in cancer patients [14].

The growing group of elderly patients, who in some cases already have a restricted quality of life (QoL), need particular attention. The elderly population with ovarian cancer generally has more comorbidities and takes several medications; however, this group tolerates the systemic therapies in a similar manner to the younger population [15,16]. The needs of this vulnerable group should be evaluated separately and paid attention to before treatment. Involving QoL questionnaires in risk assessments could increase the possibility of detecting limitations and intervening in time to improve patient outcomes.

The aim of this analysis was to evaluate the HR-QoL in patients with ovarian cancer using a PRO-based questionnaire, to compare it to the normal female population in Germany, as well as to ovarian cancer patients worldwide, and to look for further differences in our elderly patients.

## 2. Materials and Methods

This work is a subgroup analysis of the data from a prospective study ”Role of Predictive Markers for Severe Postoperative Complications in Gynecological Cancer Surgery” (RISC-GYN Trial) [12]. The study was designed to identify predictors for severe postoperative complications in patients with gynecological cancer. Ethical approval was received from the Ethics Committee of Charité (approval ID EA2/122/15), and the inclusion criteria were that patients had an age above 18 years and a histologically confirmed malignancy or a strong suspicion of a gynecologic malignancy. Written consent was obtained from all patients. From this analysis, we selected the most frequent disease in this cohort (ovarian cancer with 70%) and analyzed it separately.

Overall, 155 patients who consecutively underwent surgery for ovarian cancer were recruited prospectively from October 2015 to January 2017. The demographic data of each patient were collected prospectively. The Charlson Comorbidity Index was calculated for each patient as previously outlined in publications by Charlson et al. [17] The Eastern Cooperative Oncology Group (ECOG) performance status and the American Society of Anesthesiologists (ASA) physical status were recorded by gynecologists and anesthesiologists prior to surgery.

The European Organization for Research and Treatment of Cancer Quality of Life Questionnaire (EORTC QLQ-C30) was used in interview form to evaluate the PROs of each patient prior to surgery. The patients were asked questions by research personnel, and all survey items were documented in paper case report form (CRF) and then entered into a databank.

The EORTC QLQ-C30 Version 3.0 is a questionnaire that incorporates nine multi-item scales: five functioning scales (physical, role, cognitive, emotional, and social functioning), three symptom scales (fatigue, pain, and nausea/vomiting), and a global health and QoL scale. The remaining single items assess the additional symptoms that are commonly reported by cancer patients: dyspnea, loss of appetite, sleep disturbance, constipation, and diarrhea as well as the perceived financial effect of disease and treatment. For the five functional scales and the global QoL scale, a high score represents a good level of functioning. For the symptom scales and items, a high score corresponds to the most severe symptoms. All scale and item scores were linearly transformed to a scale from 0 to 100. The mean scores were calculated for all items of the EORTC QLQ-C30.

The first reference data were from the female population of 501 healthy women in Germany with a mean age of 54 years and extracted from a larger study conducted by the EORTC group and Nolte et al. that was published in 2019 [10]. The second reference data, which we compared to our RISC-Gyn Trial patients (155 patients with ovarian cancer), were from the EORTC QLQ-C30 Scoring Manual, based on 912 patients with ovarian cancer, that was published in 2008. This manual is only based on basic QoL data collected prior to treatment. All stages as well as relapses were included. Data from patients currently receiving advice were excluded.

Furthermore, we classified the patients from the RISC-Gyn Trial [12] into two groups: age <65 and ≥65 years, and then looked for differences in their HR-QoL according to the EORTC QLQ-C30. The groups were compared using a T-Test. The adjusted odds ratios (ORs) with a corresponding 95% confidence interval (95% CI) were attained using logistic regression analysis. For the multivariate analysis, a gradual logistic regression through data variables such as advanced-stage tumor (FIGO III–IV) and the Charlson Comorbidity Index (>2) were performed stepwise with *p*_in_ = 0.05 and *p*_out_ = 0.10. Cases with missing data were excluded from the multivariable analyses (<5%) and *p* < 0.05 was considered statistically significant. IBM^®^ SPSS^®^ Statistics 25 (SPSS Inc. an IBM Company, Chicago, IL, USA) was used for statistical analysis.

## 3. Results

Advanced-stage ovarian cancer (FIGO III–IV) was registered in 126 out of 155 patients (82.4%). Fifty-five patients enrolled in the study were ≥65 years. The patients ≥65 years had significantly more comorbidities (Charlson Comorbidity Index >2; 47% versus 19%, *p* < 0.001) and a higher rate of polypharmacy (>5 medications) (42% versus 9%, *p* < 0.001). The ASA classification >2 and ECOG >1 were more present in this group (41% versus 26%, *p* = 0.03 and 9% versus 7%, *p* = 0.01, respectively). The detailed characteristics of our patient collective are given in Table 1.

### 3.1. Comparison of the EORTC-QLQ C30 for 155 RISC-Gyn Trial Patients with Ovarian Cancer to the Female Population Norm Data for the EORTC QLQ-C30 in Germany

Except for physical functioning, all other domains of the functioning scales were significantly lower for the RISC-Gyn Trial patients with ovarian cancer than in the standard female population. Specifically, the emotional functioning and the social functioning scales showed significantly lower scores on a 100-point scale (53 points versus 74 points, *p* < 0.001 and 73 points versus 85 points, *p* > 0.001, respectively). Furthermore, the RISC-Gyn Trial patients reported more nausea and vomiting (11 points versus 8 points, *p* = 0.003) and appetite loss (23 points versus 11 points, *p* > 0.001). Insomnia was reported in both groups with significantly higher scores in the RISC-Gyn Trial patients (48 points versus 34 points, *p* < 0.001), Further differences are shown in Table 2.

### 3.2. Comparison of EORTC-QLQ C30 for 155 RISC-Gyn Trial Patients with Ovarian Cancer to the Reference Data for the QLQ-C30 Based upon Data Provided by EORTC for 917 Patients with Ovarian Cancer

Patients from the RISC-Gyn Trial scored noticeably better on the physical functioning scale than the reference group (82 points versus 76 points, *p* = 0.004). Emotional functioning was significantly lower in the RISC-Gyn Trial patients than in the reference group (53 point versus 68 points, *p* < 0.001). The symptom scales showed significant differences for pain (34 point versus 27 points, *p* = 0.007) and insomnia (48 points versus 35 points, *p* < 0.001), where worse symptoms were experienced by the RISC-Gyn Trail patients. Further differences are shown in Table 3.

### 3.3. Differences in Domains of Quality of Life in Elderly Patients with Ovarian Cancer (RISC-Gyn Trial) According to the EORTC-QLQ C30

The self-rated functioning scales in our elderly cohort were better than those reported in the younger cohort. Across subscales, the global health status was rated with 65 points for the group ≥65 years versus 55 points for the group <65 years (*p* = 0.02). Emotional (60 points versus 50 points, *p* = 0.02), cognitive (88 points versus 76 points, *p* = 0.005), and social (81 points versus 68 points, *p* = 0.006) functioning scales were higher in the elderly group. Furthermore, the elderly patients experienced significantly less fatigue (29 points versus 40 points, *p* = 0.03) and financial difficulties (2 points versus 20 points, *p* < 0.001).

### 3.4. Logistic Regression of Patient-Oriented Quality of Life Adjusted to Advanced Disease and Comorbidities

The domains of the functioning scales and symptoms, which differ in the subgroups of patients for <65 and ≥65 years, were adjusted for comorbidities (Charlson Comorbitity Index >2) and advanced-stage tumor (FIGO III–IV). The elderly group showed significantly higher scores for the global health status (*p* = 0.04). The assessment of cognitive functioning showed a benefit for the elderly group with 11 points (*p* = 0.03). The younger group reported fatigue more often, with 18 points difference in the adjusted situation (*p* = 0.001). Table 4 shows detailed information about the distribution for the groups ≥65 years and <65 years.

## 4. Discussion

The aim of this analysis was to evaluate the health-rated quality of life (HR-QoL) in patients with ovarian cancer using the patient-reported outcomes (PROs) based on the EORTC QLQ 30 questionnaire. First, we compared the cohort of 155 RISC-Gyn Trial patients with ovarian cancer to the basic data of a healthy group of women. Afterwards, we looked for differences between our cohort and the reference data from the EORTC Quality of Life Group members based on 917 patients with ovarian cancer. Finally, we analyzed specific domains of QoL in the cohort of elderly patients.

The HR-QoL for functioning scales differed from the normal female population in patients with ovarian cancer except for the physical functioning scale. There was a significant disadvantage in the emotional, role, social, and cognitive functioning and the global QoL for cancer patients. Furthermore, cancer patients suffered more frequently from nausea, vomiting, and appetite loss. Insomnia was reported in both groups, but was still significantly higher in the cancer patients. The comparisons of all groups can be found in Figure 1.

Our patients in the RISC Gyn Trial scored higher on the physical functioning scale, but lower on the emotional functioning scale, than the reference data of the EORTC group of patients with ovarian cancer. The scores for insomnia and pain were also higher in our cohort. Interestingly, the younger patients with ovarian cancer <65 years had a lower HR-QoL for the emotional, role, social, and cognitive functioning scales than the older RISC-Gyn patients. The younger patients with ovarian cancer also had higher scores for fatigue and financial difficulties.

The literature contains several publications about baseline QoL and the predictive ability for overall survival and progression-free survival, or non-surgical complications, in cancer patients [18,19]. A meta-analysis of individual patient data from EORTC clinical trials, published by Quinton et al. in 2009, showed that HR-QoL parameters, such as physical functioning, appetite loss, and pain, provide prognostic value for overall survival in addition to age, sex, performance status, and distant metastases [20].

For gynecologic cancer patients, the QoL was found to be prognostic for overall survival, progression-free survival, and chemotherapy toxicity [21,22,23,24]. Roncolato et al. demonstrated that low global health status and low scores for role and physical functioning before chemotherapy were associated with the early termination of chemotherapy in patients with ovarian cancer [25].

In our previous publications, we demonstrated that restrictions in QoL domains such as physical functioning had a significant effect on the risk of severe postoperative complications [11,12]. Furthermore, we found that patients with debilitating symptoms and cognitive impairment had an increased risk of developing severe postoperative complications [11]. Further work in our institution showed that objective and self-reported cognitive functioning, together with appetite loss, were prognostic for mortality in elderly cancer patients [26]. There are only few studies about elderly patients with ovarian cancer [27,28].

A specialized questionnaire named EORTC QLQ-ELD15 was validated for elderly cancer patients (>70 years) [29]. Unfortunately, it is barely used in the clinical routine or even in trials, where older people are already under-represented. Elderly patients have more comorbidities and use more medications. Nevertheless, we demonstrated a better self-reported QoL in the elderly patients than in the subgroup of younger patients. We also showed that they had similar compliance rates to the younger patients under chemotherapy, and that they did not have a higher risk of postoperative complications [5].

We considered the symptom scales alongside the functioning scales. An international survey of symptoms and concerns in ovarian cancer survivors done by Webber et al. analyzed 1360 patients and found that 60% of them had fatigue, 48% had mood disorders, and 59% of them suffered from insomnia [5]. Fatigue seems to be an important issue that clinicians cannot estimate objectively and that differs from patient-reported scores [20]. In our present analysis, the younger patients with ovarian cancer were more affected by fatigue and sleep disturbance. The global QoL of women with ovarian cancer neutralizes after treatment to levels like those found in the normal population, but chronic fatigue, ovarian cancer-related symptoms, neurotoxicity, depression, and sleep disturbances persist as long-term side effects in cancer survivors [30].

The measurement and awareness of HR-QoL data seems to be a crucial part of the holistic view in medicine. According to the World Health Organization, quality of life is understood as a multidimensional construct and is influenced in multiple dimensions, such as physical health, mental well-being, independence in daily life, social networking, and a healthy relationship between the individual and their environment as well as their religious and spiritual beliefs. As such, patient-reported QoL has to be understood in the context of the culture and the value system of the patient [10]. Unfortunately, the intercultural aspect is missing in our study. The intercultural aspect should be evaluated in future studies. A further limitation of the study was the use of the classical paper-form EORTC questionnaire, which was not always immediately answered by the patients, so an additional bias cannot be excluded. The evaluation of QoL using a digital QLQ-C30 computer adaptive version (CAT) questionnaire form may help to decrease this potential bias [31].

## 5. Conclusions

Cancer forces people out of their normal roles and social lives, especially younger women who are balancing work and family life. The assessment of the HR-QoL in patients in a clinical routine allows health care professionals to better evaluate their further needs. Interestingly, our patients with ovarian cancer had no significant differences in the physical functioning scale when compared with the healthy women subgroup. In contrast, our results underlined the urgent need for supportive therapy logarithms for the emotional and social areas of a patient’s daily life. In particular, the younger group would benefit from supportive professional consultations concerning work and finance management.

## Figures and Tables

**Figure 1 cancers-13-00631-f001:**
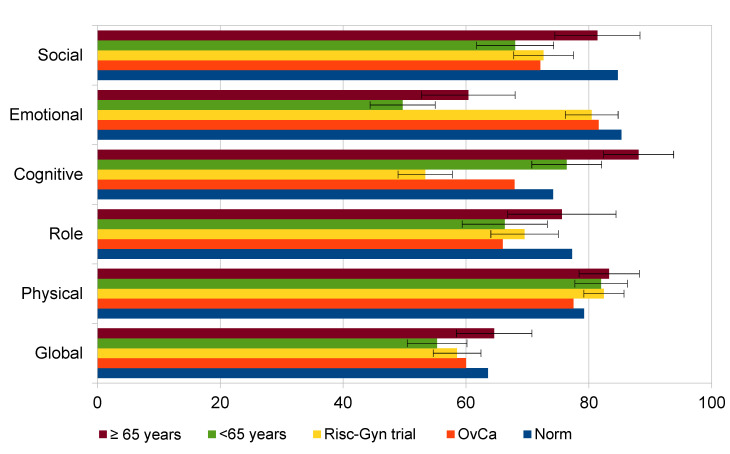
Comparison of the EORTC-QLQ C30 for 155 RISC-Gyn Trial patients with ovarian cancer (total and separated into <65 and ≥65 years) to the EORTC QLQ-C30 of a female population of 501 healthy women in Germany and the reference data from the EORTC Quality of Life Group members based on 917 patients with ovarian cancer and published in 2008.

**Table 1 cancers-13-00631-t001:** Baseline characteristics.

Baseline Characteristics	Total *n* = 155	Age <65 Years *N* = 100	Age ≥65 Years *N* = 55	*p*-Value
	(Range or %)	*n* (Range or %)	*n* (Range or %)	
ASA grade >2	48 (31.2)	26 (26)	22 (40.7)	0.03
ECOG >1	12 (7.7)	7 (7)	5 (9.1)	0.01
BMI (kg/m^2^)	24.9 (17.5–46.4)	24.2 (17.5–46.4)	26 (17.6–43.8)	0.1
Charlson Comorbidity Index >2	45 (29)	19 (19)	26 (47.3)	<0.001
Polypharmacy >5 medications	32 (20.6)	9 (9)	23 (41.8)	<0.001
FIGO Stage I–II	27 (17.6)	20 (20.4)	7 (12.7)	0.3
FIGO Stage III–IV	126 (82.4)	78 (79.6)	48 (87.3)	
Relapse Disease	49 (31.6)	36 (36)	13 (23.6)	0.2
High-grade	127 (90.8)	79 (86.8)	48 (97.9)	<0.001
Low-grade	13 (9.3)	12 (13.2)	1 (2)	
Neoadjuvant Chemotherapy	9 (5.8)	8 (8)	1 (1.8)	0.2
Presence of Ascites >500 mL	39 (25.3)	25 (25.3)	14 (25.5)	1.0
CA 125 U/mL >500	47 (32)	32 (33.3)	15 (29.4)	0.7

BMI, body mass index; ASA PS, American Society of Anesthesiologists physical status classification system; ECOG PS, Eastern Cooperative Oncology Group scale of performance status; FIGO, Fédération Internationale de Gynécologie et d’Obstétrique; CA, Cancer Antigen.

**Table 2 cancers-13-00631-t002:** Comparison of the European Organization for Research and Treatment of Cancer Quality of Life Questionnaire (EORTC-QLQ C30) for 155 RISC-Gyn Trial patients with ovarian cancer and the EORTC-QLQ C30 for the female population of 501 healthy women in Germany (the reference data from the European Organization for Research and Treatment of Cancer (EORTC) Quality of Life Group members published in 2019).

Constructed Scales	EORTC-QLQ C30 for 155 RISC-Gyn Trial Patients with Ovarian Cancer		EORTC-QLQ C30 for the Female Population of 501 Healthy Women in Germany		Difference	Significance
	Mean	SD	Mean	SD	95% Confidence Intervall	*p*-Value
Global health status	58.6	24.4	63.6	3.4	−9.0–(−1.0)	0.01
Physical functioning	82.4	20.6	79.2	5.5	−0.2–6.6	0.07
Role functioning	69.5	34.4	77.3	5.3	−13.4–(−2.1)	0.008
Emotional functioning	53.4	27.7	74.2	4.0	−25.2–(−16.5)	<0.001
Cognitive functioning	80.5	26.9	85.3	2.8	−9.2–(−0.5)	0.03
Social functioning	72.6	30.4	84.7	2.7	−16.9–(−7.2)	<0.001
Fatigue	36.6	29.6	34.0	2.2	−2.2–7.3	0.3
Nausea and vomiting	11.4	24.8	8.3	0.8	2.0–10.1	0.003
Pain	34.2	34.0	33.8	5.5	−5.3–6.2	0.9
Dyspnea	21.0	31.4	22.5	6.36	−6.6–3.7	0.6
Insomnia	47.5	38.4	33.6	6.1	7.9–20.3	<0.001
Appetite loss	22.6	33.1	10.8	0.7	6.5–17.1	<0.001
Constipation	13.3	29.1	9.5	0.7	−1.0–8.4	0.1
Diarrhea	13.3	25.8	9.5	2.6	−0.5–7.9	0.08
Financial difficulties	13.9	29.2	11.6	1.2	−2.4–6.9	0.3

*T*-test for paired samples with norm data for females in Germany (standardized for age groups).

**Table 3 cancers-13-00631-t003:** Comparison of the EORTC-QLQ C30 for 155 RISC-Gyn Trial patients with ovarian cancer and the reference data from the EORTC Quality of Life Group members based on 917 patients with ovarian cancer and published in 2008.

Constructed Scales	155 RISC-Gyn Trial Patients with Ovarian Cancer		Reference Data for the QLQ-C30 Based on Data Provided by the EORTC (917 Patients with Ovarian Cancer)		Difference	Significance
	Mean	SD	Mean	SD	95% Confidence Intervall	*p*-Value
Global health status	58.6	24.4	60.0	25.2	−5.4–2.5	0.5
Physical functioning	82.4	20.6	77.5	21.8	1.6–8.2	0.004
Role Functioning	69.5	34.4	66.0	33.5	−2.0–9.1	0.2
Emotional functioning	53.4	27.7	67.9	25.3	−19.0–(−10.1)	<0.001
Cognitive functioning	80.5	26.9	81.6	22.4	−5.5–3.2	0.6
Social functioning	72.6	30.4	72.1	31.2	− 4.4–5.4	0.8
Fatigue	36.6	29.6	37.6	28.8	− 5.8–3.7	0.7
Nausea and vomiting	11.4	24.8	11.2	21.5	− 3.8–4.2	0.9
Pain	34.2	34.0	26.7	28.7	2.1–13.0	0.007
Dyspnea	21.0	31.4	19.4	27.6	− 3.5–6.6	0.5
Insomnia	47.5	38.4	34.5	33.7	6.8–19.1	<0.001
Appetite loss	22.6	33.1	25.7	34.2	−8.4–2.2	0.3
Constipation	13.3	29.1	22.0	30.6	−13.4–(−4.1)	<0.001
Diarrhea	13.3	25.8	10.8	22.6	−1.7–6.6	0.3
Financial difficulties	13.9	29.2	13.2	26.1	−4.0–5.4	0.8

**Table 4 cancers-13-00631-t004:** Differences in domains of quality of life according to EORTC-QLQ C30 in elderly patients with ovarian cancer (RISC-Gyn Trial) and Logistic Regression for the health-related quality of life (HRQoL) for elderly patients (≥65 years adjusted to advanced-stage tumor (FIGO III–IV) and comorbidities (Charlson Comorbidity Index > 2).

Constructed Scales	55 RISC-Gyn Trial Patients ≥65 Years		100 RISC-Gyn Trial Patients <65 Years		Significance	Logistic Regression for HRQoL for Elderly (≥65 Years) Adjusted to Advanced-Stage Tumor (FIGO III–IV) and Comorbidities (Charlson Comorbidity Index >2)	
	Mean	SD	Mean	SD	*p*-Value	95% Confidence Intervall for Difference	*p*-Value
Global health status	64.6	22.82	55.3	24.70	0.02	−9.3 (−18.1–(−0.5))	0.04
Physical functioning	83.3	18.24	82.0	21.85	0.7		
Role Functioning	75.6	32.42	66.3	35.15	0.1		
Emotional functioning	60.4	28.00	49.7	26.91	0.02	−9.6 (−19.6–0.5)	0.06
Cognitive functioning	88.1	20.96	76.4	28.87	0.005	−10.8 (−20.5–(1.0))	0.03
Social functioning	81.4	25.49	68.0	31.92	0.006	−8.6 (−19.7–2.5)	0.1
Fatigue	29.3	27.13	40.4	30.30	0.03	17.8 (7.3–28.4)	0.001
Nausea and vomiting	10.3	24.29	12.0	25.20	0.7		
Pain	29.5	34.56	36.7	33.59	0.2		
Dyspnea	20.5	31.07	21.2	31.75	1.0		
Insomnia	47.4	36.96	47.5	39.29	1.0		
Appetite loss	23.9	34.82	21.9	32.35	0.7		
Constipation	14.1	30.50	12.8	28.45	0.8		
Diarrhea	12.8	24.83	13.5	26.48	0.9		
Financial difficulties	1.9	10.25	20.2	33.61	<0.001		

## Data Availability

Raw data were generated at Charité Universitätsmedizin Berlin. Data are available upon reasonable request. All data relevant to the study are included in the article.

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
