# Peer review of "Patient-Reported Outcomes (PROs) and Health-Related Quality of Life (HR-QoL) in Patients with Ovarian Cancer: What Is Different Compared to Healthy Women?"

_cancers, 2021, doi:10.3390/cancers13040631_

Round 1

Reviewer 1 Report

This work evaluates health-rated quality of life in patients with ovarian cancer using a patient-reported outcomes based questionnaire and use normal female population as control. This is an interesting work but several  revision is necessary.

Abstract must be modified so that it is better understandable

Abstract, line 21, here, the number of patients to be studied is noted but not the number of corresponding controls

Abstract, line 33, the conclusion of this analysis is not presented, it is better before discussing the results, the authors clearly concluded the results of their analysis.

In ovarian cancer, one of the major problems really impacting the social life of patients, is peritoneal metastasis and generation of carcinomatosis and ascites in the peritoneal cavity. this is not clearly identified in the study

Regarding statistical analysis, for what p -value is at the limit of significance, it is better to know the median and the max and min values for the noted scales.

Conclusion line 257; rather explains the perspective of this research work, it does not conclude the results provided by this work

Overall, this research work in the medico-social field is interesting. some  points are missing to clarify the true impact of this work. above all, depending on the stage and grade of the disease.

Author Response

Response to Reviewers

Dear Editors and Reviewers,
We would like to thank you for your useful comments to improve our paper. We appreciate the time you invested in reviewing and changed some parts of the manuscript as a result of your review. We marked the changes in red in the new manuscript file and have addressed your comments as explained below:

Comments from Reviewer and our response

This work evaluates health-rated quality of life in patients with ovarian cancer using a patient-reported outcomes based questionnaire and use normal female population as control. This is an interesting work but several revision is necessary.

Abstract must be modified so that it is better understandable

Thank you for this comment. The abstract is completely rewritten.

Abstract, line 21, here, the number of patients to be studied is noted but not the number of corresponding controls

Thank you for your remark. We have added the missing data:
HR-QoL for 155 enrolled patients with ovarian cancer was assessed by the EORTC QLQ-C30 and compared with 501 healthy women in Germany, as well as to the reference data for 917 ovarian cancer patients.

Abstract, line 33, the conclusion of this analysis is not presented, it is better before discussing the results, the authors clearly concluded the results of their analysis.

You are absolutely right. We have changed as:
Discussion: Interestingly, the patients with ovarian cancer compared with the healthy women had no significant differences in the physical functioning scale. In contrast, the patients especially the younger group need special supports for the emotional and social areas in their daily life.

Regarding statistical analysis, for what p -value is at the limit of significance, it is better to know the median and the max and min values for the noted scales.

Thank you for this comment. We have added all detailed data median, min max for all comparisons you can find the detailed figures in the revised manuscript.
with pin=0.05 and pout=0.10. Cases with missing data were excluded from the multivariable analyses (<5%). P<0.05 was considered statistically significant.

Conclusion line 257; rather explains the perspective of this research work, it does not conclude the results provided by this work

You are absolutely right, we have changed the conclusion:
Cancer throws people out of their role and pulls them out of their social life, especially for younger women who are in the middle of their work and family life. Assessment of HR-QoL in patients in clinical routine gives us the possibility to evaluate their further needs. Interestingly, our patients with ovarian cancer compared with the healthy women had no significant differences in the physical functioning scale. In contrast, our results underline the high need for supportive therapy logarithms for the emotional and social areas in patients daily life. Especially the younger group would benefit from supportive professional consultations about work and finance management.

Overall, this research work in the medico-social field is interesting. some points are missing to clarify the true impact of this work. above all, depending on the stage and grade of the disease

Thank you for this statement. You are right the stage of the disease with peritoneal carcinosis and ascites and additional other comorbidities effect the QoL of the patients, that's why we did the analysis at the table 4:Logistic Regression für HRQoL for elderly (≥65 years adjusted to advanced stage (FIGO III + IV) and Comorbidities (Charlson Comorbidity Index >2)
The domains of the functioning scales and symptoms, which differ in the subgroups of patients for <65 and ≥65 years were adjusted for comorbidities (Charlson Comorbitity Index >2) and advanced tumor stage FIGO stage III and IV.
We couldn't demonstrate it well. We added some points for better understanding.

Thank you very much for your comments. They were very comprehensible and improved our work.
Thanks, Inci

Reviewer 2 Report

The authors evaluated the health-rated quality of life (HR-QoL) in patients with ovarian cancer using a patient-reported outcomes (PROs) based questionnaire. Cancer and cancer-related treatment deeply affect QoL.
In this paper, data of 155 patients with ovarian cancer were assessed by the EORTC QLQ-C30 and compared to  data of females in Germany, as well as to the reference data for ovarian cancer based on the EORTC questionnarie.
Interestingly, the author observed that age have an importa role on QoL. Emotional, cognitive, and social functioning scales were higher in the elderly subgroup.
The paper is worthy for pubblcation, but few points deserve to be adressed.
Here, my comments:
1) As the authors stated cancer-related treatment impacts on QoL. It is interesting to underastand the exact time frame in which the patients filled the questionnaire. Pre-surgery, post-surgery and te egining or end of chemotherapy?
2) For the same reason it is interesting to understand various characteristics of patients included in the validation cohorts.   
3) This variable should be included in a moded to assess the impact of treatments on QoL
4) Please valuate all factors suggestive for low QoL values. Analysis of factors predicting for poor patient-reported outcome is needed.
5) It would be interesting to know about the changing of QoL obsrved during the patients' journey.
6) Age is one of the most important variables predicting for poor QoL, but patients aged <65 and patients aged >65 yrs, have several differences in baseline charateristics. Univariate and multivariate (if appropriate) analyses are needed to better understand this association.
7) The same paper is reported as reference with the number  15 and 22

Author Response

Response to Reviewers

Dear Editors and Reviewers,
We would like to thank you for your useful comments to improve our paper. We appreciate the time you invested in reviewing and changed some parts of the manuscript as a result of your review. We marked the changes in red in the new manuscript file and have addressed your comments as explained below:

Comments from Reviewer and our response

The authors evaluated the health-rated quality of life (HR-QoL) in patients with ovarian cancer using a patient-reported outcomes (PROs) based questionnaire. Cancer and cancer-related treatment deeply affect QoL.
In this paper, data of 155 patients with ovarian cancer were assessed by the EORTC QLQ-C30 and compared to data of females in Germany, as well as to the reference data for ovarian cancer based on the EORTC questionnarie.
Interestingly, the author observed that age have an importa role on QoL. Emotional, cognitive, and social functioning scales were higher in the elderly subgroup.
The paper is worthy for pubblcation, but few points deserve to be adressed. Here, my comments:
1) As the authors stated cancer-related treatment impacts on QoL. It is interesting to underastand the exact time frame in which the patients filled the questionnaire. Pre-surgery, post-surgery and te egining or end of chemotherapy?

Thank you for your remark. We have added the missing data:
Overall, 155 consecutive patients who underwent surgery for ovarian cancer were recruited prospectively from October 2015 to January 2017.

The European Organization for Research and Treatment of Cancer Quality of Life questionnaire (EORTC QLQ-C30) was used to evaluate PROs of each patient in an interview form prior surgery.
The first reference data was the female population of 501 healthy women in Germany substrated from the large study of the EORTC group and Nolte et al. published in 2019 [10].The second reference data, which we compared our RISC-Gyn Trial patients (155 patients with ovarian cancer) to, was the EORTC QLQ–C30 Scoring Manual based on 912 patients with ovarian cancer published in 2008. This manual is based only on basic QoL data collected prior to treatment. All stages as well as relapses were included. Data from patients currently receiving advice have been excluded.

2) For the same reason it is interesting to understand various characteristics of patients included in the validation cohorts.

Please see above. So, both groups assessed the QoL parameters before any treatment. Both cohorts include all stages and relapses. Furthermore information from the reference group is not given.

3) This variable should be included in a moded to assess the impact of treatments on QoL
You are right. Since this evaluation is a basic evaluation prior treatment we cannot measure the impact of the treatments.

4) Please valuate all factors suggestive for low QoL values. Analysis of factors predicting for poor patient-reported outcome is needed.

In table 1 you can find all demographic and clinical parameters of our patients differentiated to age. Furthermore you can find in table 4 the differences in domains of Quality of Life according to EORTC-QLQ C30 in elderly and younger patients of our cohort. Afterwards with the assumption that advanced stages and also chronic diseases could affect the quality of life, we carried out a logistic regression adjusted to advanced stage and comorbidities. We didn´t described well, so it was not clearly understood. We have changed this part and made it more clearly:
Logistic Regression für HRQoL for elderly (≥65 years adjusted to advanced stage (FIGO III + IV) and Comorbidities (Charlson Comorbidity Index >2)

5) It would be interesting to know about the changing of QoL obsrved during the patients' journey.
You are absolutely right, but unfortunately, we did the QoL assessment only before surgery. Our primary goal in the RISC Gyn Trial was to analyze QoL parameters in relation with postoperative complications. Afterwards we see the effect and the differences in QoL and did this subgroup analysis, because we thought it is worthy to know this knowledge in our clinical work.
Sehouli J, Heise K, Richter R et al. Preoperative quality of life as prediction for severe postoperative complications in gynecological cancer surgery: results of a prospective study. Arch. Gynecol. Obstet. 2020. doi:10.1007/s00404-020-05847-1.
Inci MG, Richter R, Woopen H, et al
Role of predictive markers for severe postoperative complications in gynecological cancer surgery: a prospective study (RISC-Gyn Trial)
International Journal of Gynecologic Cancer Published Online First: 27 November 2020. doi: 10.1136/ijgc-2020-001879

6) Age is one of the most important variables predicting for poor QoL, but patients aged <65 and patients aged >65 yrs, have several differences in baseline charateristics. Univariate and multivariate (if appropriate) analyses are needed to better understand this association.

Yes, you are absolutely right. I think with the response to the 4.comment and adding the logistic regression this point is already answered.

7) The same paper is reported as reference with the number 15 and 22

Round 2

Reviewer 1 Report

in the text we see some german word and can be changed to english if possible